# Synthesis and Evaluation of Two Long-Acting SSTR2 Antagonists for Radionuclide Therapy of Neuroendocrine Tumors

**DOI:** 10.3390/ph15091155

**Published:** 2022-09-16

**Authors:** Sofia Koustoulidou, Maryana Handula, Corrina de Ridder, Debra Stuurman, Savanne Beekman, Marion de Jong, Julie Nonnekens, Yann Seimbille

**Affiliations:** 1Department of Radiology and Nuclear Medicine, Erasmus MC Cancer Institute, Erasmus University Medical Center Rotterdam, 3015 GD Rotterdam, The Netherlands; 2Department of Molecular Genetics, Erasmus MC Cancer Institute, Erasmus University Medical Center Rotterdam, 3015 GD Rotterdam, The Netherlands; 3Life Sciences Division, TRIUMF, Vancouver, BC V6T 2A3, Canada

**Keywords:** neuroendocrine tumors, somatostatin receptor subtype 2, JR11, albumin binder, radionuclide therapy

## Abstract

Somatostatin receptor subtype 2 (SSTR2) has become an essential target for radionuclide therapy of neuroendocrine tumors (NETs). JR11 was introduced as a promising antagonist peptide to target SSTR2. However, due to its rapid blood clearance, a better pharmacokinetic profile is necessary for more effective treatment. Therefore, two JR11 analogs (**8a** and **8b**), each carrying an albumin binding domain, were designed to prolong the blood residence time of JR11. Both compounds were labeled with lutetium-177 and evaluated via in vitro assays, followed by in vivo SPECT/CT imaging and ex vivo biodistribution studies. [^177^Lu]Lu-**8a** and [^177^Lu]Lu-**8b** were obtained with high radiochemical purity (>97%) and demonstrated excellent stability in PBS and mouse serum (>95%). [^177^Lu]Lu-**8a** showed better affinity towards human albumin compared to [^177^Lu]Lu-**8b**. Further, **8a** and **8b** exhibited binding affinities 30- and 48-fold lower, respectively, than that of the parent peptide JR11, along with high cell uptake and low internalization rate. SPECT/CT imaging verified high tumor accumulation for [^177^Lu]Lu-**8a** and [^177^Lu]Lu-JR11 at 4, 24, 48, and 72 h post-injection, but no tumor uptake was observed for [^177^Lu]Lu-**8b**. Ex vivo biodistribution studies revealed high and increasing tumor uptake for [^177^Lu]Lu-**8a.** However, its extended blood circulation led to an unfavorable biodistribution profile for radionuclide therapy.

## 1. Introduction

Treatment of neuroendocrine tumors (NETs) largely depends on radioligands targeting somatostatin receptor type 2 (SSTR2). [^177^Lu]Lu-DOTA-TATE (Lutathera^®^) is the leading radioligand with approval from both the Food and Drug Administration (FDA) and the European Medicines Agency (EMA) [1]. The NETTER-1 phase III study showed promising results, with a response rate of 18% for the [^177^Lu]Lu-DOTA-TATE group [2]. However, novel developments are still necessary to achieve a better response. One such strategy is to enhance the radioligand delivery to increase the radiation dose to tumor cells.

Various studies have shown that SSTR2 antagonists, such as JR11, are more potent than SSTR2 agonists due to their ability to bind to more receptor binding sites. Therefore, several SSTR2 antagonist peptides have been labeled for diagnostic or therapeutic purposes [3,4,5,6,7]. However, the rapid blood clearance of such SSTR2 radioligands and the significant accumulation in non-target tissues pose a limit for higher tumor dose delivery and more efficient treatment [8]. In this context, binding of radioligands to serum protein can be an efficient method to improve the pharmacokinetic properties of these molecules [9]. Albumin binding domains (ABD) promise to increase the time-integral tumor uptake by prolonging blood circulation and reducing the uptake in healthy organs such as the kidneys [10]. Recently, Evans Blue (EB), a molecule known to bind to albumin, was conjugated to the agonist octreotate, and the resulting EB-TATE was labeled with the therapeutic radionuclide yttrium-90. [^90^Y]Y-EB-TATE showed higher tumor uptake and improved tumor response in mice bearing SSTR2-positive tumors compared to [^177^Lu]Lu-DOTA-TATE [11]. Other ABDs with different binding affinity for albumin have also been reported [12,13,14]. Of those, 4-(*p*-iodophenyl)butyryl and 4-(*p*-methoxyphenyl)butyryl were the most preferred albumin binding domains due to the aforementioned studies, in which enhanced tumor uptake and tumor-to-kidney dose ratio were observed. Thus, we report herein the synthesis of two JR11 derivatives containing one of these ABDs on the side chain of a lysine residue incorporated into the peptide sequence. Then, the in vitro characteristics of these long-acting SSTR2-antagonists (**8a** and **8b**) were evaluated. Both compounds were labeled with lutetium-177, and their in vivo distribution in tumor xenografts, overexpressing SSTR2 were investigated by SPECT/CT imaging and ex vivo biodistribution.

## 2. Results

### 2.1. Synthesis of the Long-Acting SSTR2 Antagonists

The synthesis of the two long-acting JR11 analogs **8a** and **8b** was carried out following standard Fmoc-based SPPS protocols (Figure 1). Coupling of the chelator was performed using PyBOP, as we previously observed faster reaction kinetics when using this coupling agent [15,16]. Cleavage of the peptides from the solid support resulted in the removal of most of the side chain protecting groups. However, a second treatment with neat TFA was required for the complete deprotection of the *tert*-butyl groups. Further, ivDde protection of the sidechain amino group of the lysine residue in position-3 allowed orthogonal coupling of the albumin binding domains on the additional lysine positioned between the cyclic peptide and the chelator. Conjugation of the ABDs to **6** was performed after their activation as NHS esters. The final products **8a** and **8b** were obtained at 9.9 and 8.7% yield, respectively, after deprotection of the ivDde protecting group of Lys^3^ and purification by HPLC (Appendix A).

### 2.2. Radiolabeling with [^177^Lu]LuCl_3_

Labeling of **8a** and **8b** with [^177^Lu]LuCl_3_ was performed in the presence of gentisic acid and ascorbic acid. Both additives are known scavengers to protect radiopharmaceuticals from radiolytic degradation [17]. Kolliphor, a non-ionic solubilizer and emulsifier used to improve the solubility of hydrophobic compounds, was added to the labeling reaction mixture to prevent stickiness of the peptides. The radiochemical yield (RCY) of both peptides and their radiochemical purity (RCP) are presented in Table 1. [^177^Lu]Lu-**8a** and [^177^Lu]Lu-**8b** were obtained in 97.7 and 98.2% RCP, respectively (Appendix A). Both tracers showed excellent stability in phosphate buffered saline (PBS) and mouse serum up to 24 h at 37 °C (Appendix A). The radioligands exhibited negative LogD_7.4_ values, and [^177^Lu]Lu-**8a** bound more efficiently to human albumin than did [^177^Lu]Lu-**8b** (Table 1).

### 2.3. Competitive Binding Assay

The IC_50_ values of **8a** and **8b** were determined in a competitive binding assay using U2OS-SSTR2 cells and [^177^Lu]Lu-JR11 as radioligand (Table 2 and Appendix A); **8a** and **8b** exhibited a binding affinity of 80 and 130 nM, respectively, which is 30- and 48-fold lower than the IC_50_ value found for the parent peptide JR11. Considering that the binding affinities of **8a** and **8b** were still in the nanomolar range, we considered that they were suitable for further studies.

### 2.4. Uptake and Internalization in U2OS-SSTR2 Cells and H69 Tumor Tissues

Uptake of ^177^Lu-labeled **8a** and **8b** was observed in an in vitro assay using U2OS-SSTR2 cells (7.8 ± 0.05 and 3.1 ± 0.33% added dose, respectively (Figure 1A). Most radioactivity uptake was membrane-bound in both cases (5.8 ± 0.17 and 2.6 ± 0.38% added dose, respectively), confirming the antagonist properties of the two compounds. However, compared to the [^177^Lu]Lu-JR11 reference (total uptake 16.2 ± 3.0% added dose), this uptake was significantly lower, especially for [^177^Lu]Lu-**8b**. A similar radioactive uptake pattern (930 ± 37 DLU/mm^2^ for [^177^Lu]Lu-JR11, 561 ± 18 DLU/mm^2^ for [^177^Lu]Lu-**8a**, and 280 ± 23 DLU/mm^2^ for [^177^Lu]Lu-**8b**) was observed when H69 human carcinoma tissues were incubated with the aforementioned compounds, as determined by autoradiography (Figure 1B).

### 2.5. In Vivo Evaluation by SPECT/CT Imaging

The biodistribution of lutetium-177-labeled **8a** and **8b** was first evaluated in vivo by single-photon emission computed tomography/computed tomography (SPECT/CT) imaging (Figure 2, Table 3). Mice bearing SSTR2-positive H69 xenografts were scanned at 4, 24, 48, and 72 h after intravenous (i.v.) injection of [^177^Lu]Lu-**8a** or [^177^Lu]Lu-**8b**. [^177^Lu]Lu-JR11 was used as a positive control. The maximum intensity projections (MIP) depicted in Figure 2A showed high uptake of [^177^Lu]Lu-**8a** in the tumor at 4, 24, 48, and 72 h post-injection (3.7 ± 0.9, 4.9 ± 0.5, 5.7 ± 0.6, and 4.9 ± 0.3% ID/mL, respectively) and substantial uptake in the heart. In contrast, [^177^Lu]Lu-**8b** showed no tumor uptake, especially at later time points (0.8 ± 0.1 and 0.6 ± 0.1% ID/mL at 48 and 72 h, respectively), and substantial accumulation in the kidneys (4.3 ± 0.3 and 3.1 ± 0.5% ID/mL at 48 and 72 h, respectively) (Figure 2B). Tumor uptake of the reference compound [^177^Lu]Lu-JR11 could be observed until 24 h post-injection (3.3 ± 0.7% ID/mL). High kidney uptake was observed for all three compounds, probably as a result of urinary excretion. Based on the imaging data, we determined that [^177^Lu]Lu-**8a** was suitable for further ex vivo biodistribution studies.

### 2.6. Ex Vivo Biodistribution Analysis

Next to the in vivo SPECT/CT imaging studies, ex vivo biodistribution was carried out after administration of [^177^Lu]Lu-JR11 and [^177^Lu]Lu-**8a** at the same time points (4, 24, 48, and 72 h). Compound **8b** was excluded due to the minimal tumor uptake observed earlier in the in vivo SPECT scans. An additional group of mice per compound received an excess of the corresponding non-radioactive compound (block group) to determine receptor specificity. At 4 h post-injection, the uptake of [^177^Lu]Lu-**8a** in the blood was at 21.3 ± 1.1% ID/g and gradually decreased to 8.6 ± 0.5% ID/g at 72 h post-injection, reflecting the albumin-binding properties of the compound (Figure 3A and Appendix A). Tumor uptake of [^177^Lu]Lu-**8a** slightly increased over time (4.1 ± 0.5% ID/g at 4 h, 6.5 ± 0.2% ID/g at 24 h, 7.3 ± 0.4% ID/g at 48 h, and 7.3 ± 0.9% ID/g at 72 h). However, the kidneys showed high uptake at the early and late time points (9.6 ± 1.2% ID/g at 4 h and 21.6 ± 1.4% ID/g at 72 h), suggesting clearance of the compound through renal excretion. Shortly after injection, a substantial level of radioactivity was also found in the lungs, pancreas, skin, and spleen. Finally, only a slight reduction was observed in the tumor after injection of the blocking agent.

In contrast, the reference compound [^177^Lu]Lu-JR11 showed high tumor uptake at 4 h post-injection (8.4 ± 0.5% ID/g, Figure 3B and Appendix A) and even though the tumor uptake slightly decreased over time (6.1 ± 0.5% ID/g at 24 h, 4.6 ± 0.3% ID/g at 48 h, and 3.6 ± 0.4% ID/g at 72 h post-injection), the tumor-to-kidney ratio increased from 0.6 ± 0.05 at 4 h post-injection to 1.2 ± 0.3 at 72 h post-injection. Complete blocking of tumor uptake (Block group) at 24 h post-injection confirmed the specificity of the JR11 compound for the SSTR2 receptor.

## 3. Discussion

Somatostatin receptor subtype 2 (SSTR2) is present at a high incidence in neuroendocrine tumors (NETs) and therefore is an ideal target for imaging and therapy of this malignant disease. Somatostatin analogs have been widely used for the past decades and are considered a gold standard for NET treatment [19]. Among these drugs, radiolabeled somatostatin antagonists such as LM3, JR10, and JR11 have shown greater promise than agonists (e.g., DOTA-TATE and DOTA-TOC) [20]. However, preclinical and clinical studies have demonstrated that JR11 is cleared rapidly from the bloodstream, thus leading to high kidney uptake [7,21]. Therefore, our study aimed to develop two JR11 analogs carrying two different albumin binding domains to improve the pharmacokinetic profile of JR11.

The chemical structure of our new ligands (**8a** and **8b**) is directly based on the structure of the parent peptide JR11. The introduction of the albumin binding domains into the peptide sequence was established by the attachment of a lysine residue between the cyclic peptide and the DOTA chelator. This method has been previously employed to introduce fluorescent dye on the chemical structure of existing radioligands [22,23,24]. Here, we investigated the influence of two ABDs, namely 4-(*p*-iodophenyl)butyryl and 4-(*p*-methoxyphenyl)butyryl, on the in vivo behavior of JR11. These ABDs were selected due to the promising results reported in previous studies [12,13,14,25]. The 4-(*p*-iodophenyl)butyryl group has been widely used to improve the bioavailability of different radioligands, such as DOTA-TATE, PSMA-617, and folic acid. However, when conjugated to PSMA-617 ([^177^Lu]Lu-HTK01169), Hsiou-Ting Kuo et al. noticed that not only was tumor uptake 8.3-fold higher for [^177^Lu]Lu-HTK01169 in comparison to [^177^Lu]Lu-PSMA-617, but also, the absorbed dose in the kidneys was 17.1-fold higher than that of the parent molecule [26]. Later, the same group reported a study comparing several albumin binding domains and concluded that 4-(*p*-methoxyphenyl)butyryl could be a potential candidate to improve blood circulation and reduce kidney radiotoxicity [13]. Radiolabeling of **8a** and **8b** with [^177^Lu]LuCl_3_ was successful, and both radiopeptides showed high RCYs and RCPs. They exhibited excellent inertness in PBS and mouse serum up to 24 h post-incubation, proving their stability towards radiolysis and peptidase digestion. When compared to JR11, the LogD_7.4_ values of our radiopeptides were slightly higher than the LogD_7.4_ value of the parent peptide [18]. This is probably due to the lipophilic character of the two albumin binding domains. This observation is also confirmed by the data provided by Hsiou-Ting Kuo et al. [13]. Nevertheless, the LogD_7.4_ values of [^177^Lu]Lu-**8a** and [^177^Lu]Lu-**8b** remained negative, suggesting that they are hydrophilic. [^177^Lu]Lu-**8a** showed good binding to plasma proteins, as previously reported for other radioligands bearing the same ABD [27,28]. Furthermore, [^177^Lu]Lu-**8a** showed stronger affinity towards human albumin in comparison to that of [^177^Lu]Lu-**8b**, confirming that the interaction between the ABD and the plasma proteins is influenced by the lipophilicity of the substituted phenyl group [13]. Thus, two JR11 analogs were prepared and, as expected, **8a** interacted strongly with albumin, while **8b** exhibited a milder interaction with albumin. The parent peptide JR11 showed lower binding to plasma proteins in comparison to that of our long-acting analogs.

Competitive binding assays were performed to determine the IC_50_ values of the newly synthesized compounds **8a** and **8b** for the SSTR2 receptor. The incorporation of the albumin binding moiety did affect the SSTR2 binding affinity of the two compounds, probably due to the sensitivity of the parent peptide JR11 to chemical modifications. Fani et al. noticed that different chelators affect the binding affinity of the peptide in vitro [7]. More specifically, exchanging the DOTA chelator with NODAGA ([^68^Ga]Ga-NODAGA-JR11) significantly increased the affinity of the peptide toward the SSTR2 receptor. The authors speculated that in the case of the DOTA chelator, the geometry is hexacoordinate, and in solution, the chelator can act as a spacer, hence lowering the affinity. In vitro uptake of these compounds in cells and tumor sections expressing SSTR2 was lower than the uptake of JR11. The majority of the uptake was found in the membrane-bound fraction, confirming the compounds’ antagonist properties.

SPECT/CT images revealed increased blood residence time of the compound [^177^Lu]Lu-**8a** compared with [^177^Lu]Lu-JR11. This suggests that the albumin binding affinity contributed to the different pharmacokinetic profiles. However, despite the higher tumor uptake, as observed by image analysis, accumulation in the kidneys was substantial, thus making it unsuitable for future therapeutic studies. In a different study, Rousseau et al. also observed the same pattern when comparing DOTA-TATE with an albumin binding moiety ([^177^Lu]Lu-AspAB-DOTA-TATE) to the reference analog, highlighting that albumin binding moieties might negatively affect the pharmacokinetic profile of peptides [25]. In the clinic, the high kidney uptake can be reduced by perfusion of cationic amino acids, but bone marrow toxicity will remain a problem [29]. Unfortunately, and to our surprise, compound [^177^Lu]Lu-**8b** showed no tumor uptake and rapid renal clearance despite earlier reports by Kuo et al. showing the opposite effect with PSMA ligands [13].

Ex vivo biodistribution analysis confirmed the uptake pattern observed during imaging. In addition, the introduction of a blocking group for both **8a** and JR11 showed non-specific tumor uptake for [^177^Lu]Lu-**8a** compared to [^177^Lu]Lu-JR11, for which a total blockade of SSTR2 receptor uptake was achieved. The high uptake of the compound in most organs could suggest that the plasma protein-binding prolongs the blood circulation substantially and interferes with the binding of **8a** to tumor cells, thus causing inefficient blockade. Müller and coworkers also used the 4-(*p*-iodophenyl)butyryl ABD conjugated to a folate radioligand, and they observed higher tumor uptake and a reduction in kidney accumulation [14]. However, the authors did not perform a blocking study, making it difficult to draw any conclusions on the specificity of their compound. On the other hand, van Tiel et al. did not observe total blockage of tumor uptake when they conjugated the same albumin binding domain onto Albutate-1 [12]. More specifically, the tumor uptake of [^177^Lu]Lu-Albutate-1 was 24.42 ± 1.44% ID/g at 24 h post-injection, while the blocked group showed uptake of approximately 12% ID/g. From dosimetry calculations, the authors noticed a high radiation dose in several tissues, especially in bone marrow (total absorbed dose of 765 mGy/MBq), probably due to the prolonged circulation. Even though in this study bone marrow uptake was negligible due to the inadequate extraction of the sample during the ex vivo biodistribution, this could pose a limit for a better therapeutic index with compounds carrying albumin binding moieties.

In a more recent study, the 4-(*p*-iodophenyl)butyryl moiety was conjugated to the DOTA-(PEG_28_)_2_-A20FMDV2 peptide labeled with [^177^Lu]Lu and used for the imaging of αvβ6-positive tumors [30]. The authors noticed that even though blood uptake of the peptide was high at 1 h post-injection, it dropped rapidly by 48 h (0.04 ± 0.01% ID/g), while tumor uptake remained constant (4.06 ± 0.54% ID/g), rendering the tumor-to-blood ratio ideal for therapy. Kidney uptake was, however, responsible for the toxicity observed during these studies. Based on this, it is recommended to perform the blocking study at a later timepoint to allow for better clearance of the peptide from the blood.

Overall, the previously mentioned studies, together with our results, point out the need for different and better albumin binding moieties and further modifications to the peptide itself that will improve the binding affinity and the specificity of SSTR2-targeting antagonists towards the receptor.

## 4. Materials and Methods

### 4.1. General Information

The chemicals and solvents mentioned in this manuscript were purchased from commercial suppliers and used without further purification. Fmoc-based solid-phase peptide synthesis (SPPS) of the peptide was performed manually using dedicated reaction vessels (Chemglass, Vineland, NJ, USA). DOTA-tris(*t*Bu)ester was purchased from Macrocyclics (Plano, TX, USA). Lutetium-177 (LuMark^®^ Lutetium-177 chloride) was purchased from IDB Holland (Baarle-Nassau, The Netherlands). High-performance liquid chromatography (HPLC) and mass spectrometry (MS) were carried out on an LC/MS 1260 Infinity II system from Agilent (Middelburg, The Netherlands). Analyses were performed on an analytical column (Poroshell 120, EC-C18, 2.7 μm, 3.0 × 100 mm) from Agilent with a gradient elution of acetonitrile (ACN) (5% to 100% in H_2_O, containing 0.1% formic acid) at a flow rate of 0.5 mL/min over 8 min. Nuclear magnetic resonance (NMR) spectra were recorded in deuterated dimethyl sulfoxide (DMSO-d6) and chloroform-d on a Bruker AVANCE 400 (Leiden, The Netherlands) or a Nanalysis 60 Pro (Calgary, AB, Canada) at ambient temperature. Chemical shifts are given as *δ* values in ppm, and coupling constants *J* are given in Hz. The splitting patterns are reported as s (singlet), d (doublet), t (triplet), q (quadruplet), qt (quintuplet), m (multiplet), and br (broad signal). Purification of AB2-NHS ester and compound **8b** was carried out on a preparative HPLC 1260 Infinity II system from Agilent (Middelburg, The Netherlands) using a preparative column (50 × 21.2 mm, 5 μm) from Agilent. AB2-NHS ester was purified using a gradient elution of ACN (10% to 95% in H_2_O, containing 0.1% formic acid (FA)) at a flow rate of 10 mL/min over 10 min. Compound **8b** was purified via an isocratic elution of ACN (25% in H_2_O) at a flow rate of 10 mL/min over 10 min. Purification of compound **8a** was performed on a semi-preparative HPLC e2695 Separation Module from Waters (Etten-Leur, The Netherlands) using a semi-preparative C18 Luna^®^ column (250.0 × 10.0 mm, 5 μm) from Phenomenex (Torrance, CA, USA) with an isocratic elution of ACN (35% in H_2_O) at a flowrate of 3 mL/min over 30 min. Instant thin-layer chromatography (iTLC-SG) plates on silica-gel-impregnated glass fiber sheets (Agilent; Folsom, CA, USA) were eluted with sodium citrate (0.1 M, pH 5). The plates were analyzed by a bSCAN radio-chromatography scanner from Brightspec (Antwerp, Belgium) equipped with a sodium iodide detector. The radioactive samples used to determine LogD_7.4_ in in vitro assays and in vivo studies were counted using a Wizard 2480 gamma counter (Perkin Elmer; Waltham, MA, USA). Activity measurements were performed using a VDC-405 dose calibrator (Comecer; Joure, The Netherlands). Quality control of the radiolabeled compounds and analysis of their stability were carried out on an ultra-high performance liquid chromatography (UHPLC) Acquity Arc system from Waters (Etten-Leur, The Netherlands) equipped with a diode array detector, a radio-detector from Canberra (Zelik, Belgium), and an analytical C18 Gemini^®^ column (Phenomenex; 250.0 × 4.6 mm, 5 μm) eluted with a gradient of ACN (5 to 95% in H_2_O, containing 0.1% trifluoroacetic acid (TFA)) at a flowrate of 1 mL/min over 30 min.

### 4.2. Chemistry

#### 4.2.1. Synthesis of DOTA-Lys-Phe(4-Cl)-c[D-Cys-Aph(Hor)-D-Aph(Cbm)-Lys(ivDde)-Thr-Cys]-D-Tyr-NH_2_ (**6**)

The peptide sequence was synthesized using the standard *N^α^*-Fmoc solid-phase peptide synthesis (SPPS) strategy. Peptide synthesis started by loading Fmoc-D-Tyr(*t*Bu)-OH (0.5 mmol, 2 equiv.) onto the Rink amide MBHA resin (370 mg, average loading capacity: 0.678 mmol/g). The resin was stirred for 2 h at room temperature (rt). Then, the resin was capped using a mixture of acetic anhydride (50 equiv.) and DIPEA (50 equiv.) for 1 h at rt. Elongation of the peptidyl-resin was carried out in dimethylformamide (DMF) and in the presence of *N*-[(dimethylamino)-1*H*-1,2,3-triazolo-[4,5-*b*]pyridine-1-ylmethylen]-*N*-methylmethanaminium hexafluorophosphate *N*-oxide (HATU) and *N*,*N*-diisopropylethylamine (DIPEA). Fmoc deprotection was accomplished by treatment of the peptide with a 20% solution of piperidine in DMF. Subsequent couplings/Fmoc deprotections were performed with the following protected amino acids (2 equiv.): Fmoc-Cys(Acm)-OH, Fmoc-L-Thr(*t*Bu)-OH, Fmoc-L-Lys(ivDde)-OH, Fmoc-D-Aph(*t*Bu-Cbm)-OH, Fmoc-Aph(Hor)-OH, Fmoc-D-Cys(Acm)-OH, and Fmoc-Phe(4-Cl)-OH. Amide formation and Fmoc deprotection were monitored by Kaiser and TNBS tests. Double coupling and deprotection were performed when the reactions were not complete to obtain resin-bound Phe(4-Cl)-D-Cys(Acm)-Aph(Hor)-D-Aph(*t*Bu-Cbm)-Lys(ivDde)-Thr(*t*Bu)-Cys(Acm)-D-Tyr(*t*Bu) (**1**). Coupling of Fmoc-L-Lys(Boc)-OH (2 equiv.) to **1** according to the protocol described above provided **2**. Cyclization of the peptide was performed by treatment of **2** with thallium (III) trifluoroacetate (Tl(TFA)_3_) (2 equiv.) in DMF for 1 h at rt. The cyclization reaction was monitored by LC/MS after cleavage of a small sample of peptide from the solid support. Upon Fmoc deprotection, compound **5** was obtained by coupling of DOTA-tris (*t*Bu) ester (3 equiv.) to **4** using benzotriazole-1-yl-oxy-tris-pyrrolidino-phosphonium hexafluorophosphate (PyBOP) (3 equiv.) and DIPEA (6 equiv.) in DMF for 2 h at rt. Finally, cleavage and removal of the sidechain protecting groups were performed by reacting **5** with a solution of trifluoroacetic acid/triisopropylsilane/water (TFA/TIS/H_2_O) (2 mL, v:v:v = 95:2.5:2.5) for 2 h at rt. The resin was washed twice with the cleavage cocktail, and after evaporation of the solvent, the peptide was treated with neat TFA to completely remove the *tert*-butyl groups. Solvent was evaporated with gentle airflow, and the product was precipitated in cold diethyl ether and collected by centrifugation to obtain **6** as a yellowish solid (46.3, 29%). Analytical HPLC retention time of **6 was**: *t*_R_ = 4.08 min; ESI-MS: *m*/*z*, calculated: 2021.9, found: 1012.7 [M + 2H]^2+^.

#### 4.2.2. Synthesis of 2,5-dioxopyrrolidin-1-yl 4-(4-iodophenyl)butanoate (AB1-NHS Ester)

The 4-(*p*-iodophenyl)butanoic acid (500 mg, 1.7 mmol) and *N*-hydroxysuccinimide (NHS) (198 mg, 1.7 mmol, 1 equiv.) were dissolved in anhydrous tetrahydrofuran (THF) (10 mL) under nitrogen atmosphere. *N*,*N′*-Dicyclohexylcarbodiimide (DCC) (354 mg, 1.7 mmol, 1 equiv.) was dissolved in anhydrous THF (4 mL) under nitrogen atmosphere and added dropwise to the reaction mixture. The reaction mixture was stirred for 1 h at 0 °C and left to react overnight at rt. The white precipitate was removed by filtration, and the solvent was evaporated under vacuum. The crude compound was purified by flash chromatography (hexane/ethyl acetate 4:1 → 1:1) to yield AB1-NHS ester as white crystals (492 mg, 75%). Analytical HPLC retention time of AB1-NHS ester was: *t_R_* = 6.0 min; purity > 90%; 1H NMR (400 MHz, DMSO-d6): δ 7.65 (d, *J* = 7.8 Hz, 2H), 7.05 (d, *J* = 7.9 Hz, 2H), 2.82 (s, 4H), 2.64 (dt, *J* = 12.0, 7.6 Hz, 4H), 1.94–1.84 (m, 2H).

#### 4.2.3. Synthesis of 2,5-dioxopyrrolidin-1-yl 4-(p-methoxyphenyl)butanoate (AB2-NHS Ester)

AB2-NHS ester was obtained by reacting 4-(4-methoxyphenyl)butanoic acid (500 mg, 2.6 mmol) with NHS (296 mg, 2.6 mmol, 1 equiv.) and DCC (530 mg, 2.6 mmol, 1 equiv.) following the protocol described above for AB1-NHS ester. The crude compound was purified by prep-HPLC, and AB2-NHS ester was obtained as a white solid (564 mg, 75%). Analytical HPLC retention time of AB2-NHS ester was: *t_R_* = 5.5 min; purity > 95%; 1H NMR (60 MHz, Chloroform-d) δ 7.23–6.71 (m, 4H), 3.79 (s, 3H), 2.83 (s, 4H), 2.59 (t, *J* = 6.3 Hz, 4H), 2.29–1.81 (m, 2H).

#### 4.2.4. Synthesis of DOTA-Lys(4-(p-iodophenyl)butyryl)-Phe(4-Cl)-c[D-Cys-Aph(Hor)-D-Aph(Cbm)-Lys(ivDde)-Thr-Cys]-D-Tyr-NH_2_ (**7a**)

Compound **7a** was obtained by adding AB1-NHS ester and triethylamine (Et_3_N) to peptide **6** (57 mg, 28 µmol) dissolved in 1 mL of a mixture of H_2_O/ACN (v:v = 1:1). Et_3_N (39 µL, 0.28 mmol, 10 equiv.) was added to reach pH 9, followed by the addition of AB1-NHS ester (16.4 mg, 42 µmol, 1.5 equiv.). The reaction mixture was stirred at rt, and progress of the reaction was monitored by LC/MS. Solvent was evaporated under reduced pressure, and the crude product **7a** was obtained as a yellowish solid (82 mg, 68%). Analytical HPLC retention time of **7a** was: *t_R_* = 15.7 min; ESI-MS: m/z, calculated: 2293.8, found: 1148.8 [M + 2H]^2+^.

#### 4.2.5. Synthesis of DOTA-Lys(4-(p-methoxyphenyl)butyryl)-Phe(4-Cl)-c[D-Cys-Aph(Hor)-D-Aph(Cbm)-Lys(ivDde)-Thr-Cys]-D-Tyr-NH_2_ (**7b**)

Compound **7b** was obtained by following the protocol described above for **7a** and starting from **6** (28.9 mg, 14 µmol) and AB2-NHS ester (6.24 mg, 21 μmol, 1.5 equiv.). Crude product **7b** was obtained as a yellowish solid (36.2 mg, 69%). Analytical HPLC retention time of **7b** was: *t_R_* = 4.55 min. ESI-MS: *m*/*z*, calculated: 2197.9, found: 1100.6 [M + 2H]^2+^.

#### 4.2.6. Synthesis of DOTA-Lys(4-(p-iodophenyl)butyryl)-Phe(4-Cl)-c[D-Cys-Aph(Hor)-D-Aph(Cbm)-Lys-Thr-Cys]-D-Tyr-NH_2_ (**8a**) and DOTA-Lys(4-(p-methoxyphenyl)butyryl)-Phe(4-Cl)-c[D-Cys-Aph(Hor)-D-Aph(Cbm)-Lys-Thr-Cys]-D-Tyr-NH_2_ (**8b**)

The ivDde deprotection was performed by treatment of **7a** and **7b** with a solution of 2% hydrazine in DMF. The reaction mixture was stirred for 1 h at rt. Progress of the reaction was analyzed by LC/MS. Then, the reaction mixture was diluted in H_2_O/ACN (v:v = 1:1) and purified by semi-preparative HPLC for compound **8a** and preparative HPLC for compound **8b**. Compound **8a** was obtained as a white solid (13 mg, 9.9%) Analytical HPLC retention time of **8a** was: *t_R_* = 4.0 min; purity > 96% (Appendix A); ESI-MS: *m*/*z*, calculated: 2087.7, found: 1045.5 [M + 2H]^2+^ and 697.2 [M + 3H]^3+^ (Appendix A). Compound **8b** was obtained as a white solid (7.1 mg, 8.7%) Analytical HPLC retention time of **8b** was: *t_R_* = 3.8 min; purity > 94% (Appendix A); ESI-MS: *m*/*z*, calculated: 1991.8, found: 997.6 [M + 2H]^2+^ and 665.3 [M + 3H]^3+^ (Appendix A).

### 4.3. Radiochemistry

#### 4.3.1. Lutetium-177 Radiolabeling of **8a** and **8b**

[^177^Lu]LuCl_3_ was obtained as a 0.05 M hydrochloric acid (HCl) aqueous solution. A total of 100 MBq was added to **8a** or **8b** (1 nmol), ascorbic/gentisic acids (10 μL, 50 mM), sodium acetate (1 μL, 2.5 M), and kolliphor in H_2_O (2.0 mg/mL, 60.8 μL). The mixture was incubated for 20 min at 90 °C and then left to cool down for 5 min. The radiolabeling yield was determined by instant thin-layer chromatography on silica-gel-impregnated glass fiber sheets eluted using a solution of sodium citrate (0.1 M, pH 5). Diethylenetriaminepentaacetic acid (DTPA, 5 μL, 4 mM) was added to complex free Lu-177. The radiochemical purity of [^177^Lu]Lu-**8a** and [^177^Lu]Lu-**8b** was determined by radio-HPLC (Appendix A).

#### 4.3.2. Determination of Lipophilicity

The distribution coefficient (LogD_7.4_) of the ^177^Lu-labeled peptides was determined by the shake-flask method. For each radioligand, the experiment was performed in triplicate. The radiolabeled compound (~1.5 MBq) was added to a solution of phosphate buffered saline (PBS)/n-octanol (1 mL, v:v = 1:1) in eppendorf vials. The vials were vortexed vigorously and centrifuged at 10,000 rpm for 3 min. The n-octanol phase was separated from the aqueous phase, poured into new vials, and centrifuged for 15 min. Samples from each phase (10 μL) were measured using a gamma counter. The LogD_7.4_ value was calculated using the following equation: LogD_7.4_ = ([counts in the n-octanol phase]/[counts in the PBS phase]).

#### 4.3.3. Stability Studies in PBS and Mouse Serum

The ^177^Lu-labeled compounds (~3 MBq) were incubated in PBS (300 μL) at 37 °C. The stability of the radiolabeled peptides was verified by radio-HPLC at 1, 4, and 24 h. The stability in serum was determined by incubating the radiolabeled compounds (~3 MBq) into 150 μL of mouse serum (Merck; Haarlerbergweg, The Netherlands) at 37 °C. At 1, 4, and 24 h post-incubation, the proteins were precipitated by adding an aliquot of 35 μL of the mixture to an equal volume of ACN. The vial was vortexed and centrifuged for 20 min. stability was monitored by radio-HPLC (Appendix A).

#### 4.3.4. Albumin Binding Properties

Compounds **8a**, **8b**, and JR11 were radiolabeled with lutetium-177 at a molar activity of 50 MBq/nmol. Radiopeptides (~1 MBq) were incubated in either PBS (500 μL) or human albumin/PBS (500 µL, v:v = 1:4) for 1 h at 37 °C. Three aliquots of 10 μL were counted in a gamma counter to determine the amount of activity in the loading solution. The mixtures were loaded onto Centrifree Ultrafiltration devices (Centrifree Ultrafiltration device with Ultracel PL membrane, 30 KDa, Merck, Haarlerbergweg, The Netherlands) preconditioned with 700 μL of kolliphor in PBS (0.06 mg/mL). The Centrifree Ultrafiltration devices were centrifuged at 7000 rpm for 30 min. Three aliquots of 10 μL from each mixture were counted in a gamma counter, and the protein-bound fraction was calculated based on the radioactivity measured in the filtrate relative to the corresponding loading solution.

### 4.4. In Vitro Assays

#### 4.4.1. Cell Lines and Culture

Human osteosarcoma cells (U2OS) stably expressing the SSTR2 receptor were used in all in vitro cell assays [21]. Cells were cultured in Dulbecco’s modified Eagle’s medium (DMEM) from Gibco (Paisley, UK) supplemented with 2 mM L-glutamine, 10% fetal bovine serum (FBS), 50 units/mL penicillin, and 50 μg/mL streptomycin (Sigma Aldrich; Haarlerbergweg, The Netherlands) and maintained at 37 °C and in a 5% CO_2_ humidified chamber. Passages were performed weekly using trypsin/EDTA (0.05%/0.02% *w*/*v*).

#### 4.4.2. Competitive Binding Assay

Competitive binding experiments against [^177^Lu]Lu-JR11 were performed with **8a** and **8b** in U2OS-SSTR2 cells. Cells were seeded in a 24-well plate 24 h in advance (2 × 10^5^ cells/well). On the day of the experiment, medium was removed, and the cells were washed once with PBS (Gibco). Then, solutions containing unlabeled compound **8a** or **8b** in increasing concentrations (10^−12^ to 10^−5^ M) in internalization medium (DMEM media, 20 mM HEPES, 1% BSA, pH 7.4) were added, followed by the addition of [^177^Lu]Lu-JR11 (10^−9^ M). For each concentration, experiments were performed in triplicate. Cells were incubated at 37 °C for 90 min. After incubation, medium was removed, and cells were washed once with PBS and lysed with 0.5 M sodium hydroxide (NaOH) for 10 min at rt. The lysate was transferred to counting tubes, and measurement was performed using the *γ*-counter.

#### 4.4.3. Uptake and Internalization Assay

Cells were seeded in 6-well plates 48 h before the experiment (2 × 10^5^ cells/well). The following day, adhered cells were incubated with 10^−9^ M of [^177^Lu]Lu-JR11, [^177^Lu]Lu-**8a**, or [^177^Lu]Lu-**8b** in 1 mL of culture medium for 2 h at 37 °C. After incubation, medium was removed, and cells were washed twice with PBS. The membrane-bound fraction was collected by incubating cells with an acid buffer (50 mM glycine, 100 mM sodium chloride (NaCl), pH 2.8) for 10 min at rt. Cells were lysed using 0.5 M NaOH for 10 min at rt to acquire the internalized fraction. Both fractions were counted in a *γ*-counter, and data were expressed as percentage of added dose.

#### 4.4.4. Uptake and Autoradiography of H69 Tumor Sections

Subcutaneous fresh frozen H69 tumor tissues were cut at 10 μm thickness and immediately mounted on Starfrost glass slides (Thermo Scientific; Bleiswijk, The Netherlands). Tissue sections were incubated with washing buffer (167 mM Tris-HCl, 5 mM MgCl_2_) containing 0.25% BSA for 10 min at rt to prevent nonspecific binding. Then, each section was incubated with 10^−9^ M of [^177^Lu]Lu-JR11, [^177^Lu]Lu-**8a** or [^177^Lu]Lu-**8b** diluted in incubation buffer (washing buffer containing 1% BSA) for 90 min at rt. Each slide was drained off and washed with PBS. Finally, the slides were exposed to super-resolution phosphor screens for 48 h and imaged with the Cyclone system (PerkinElmer; Waltham, MA, USA). Images were analyzed and quantified using the Optiquant software Version 5 (PerkinElmer; Waltham, MA, USA).

### 4.5. In Vivo Studies

All animal experiments were approved by the Animal Welfare Committee of the Erasmus MC, and all procedures were conducted according to accepted guidelines. Mice were subcutaneously inoculated with 5 × 10^6^ SSTR2-positive H69 human small cell lung carcinoma cells in Matrigel, and tumors were left to grow for 3–4 weeks to an average volume of approximately 300 mm^3^. When the tumors reached the desired volume, each animal was injected intravenously (i.v.) through the tail vein with 100 μL of 20 MBq/0.5 nmol [^177^Lu]Lu-JR11, [^177^Lu]Lu-**8a**, or [^177^Lu]Lu-**8b** diluted in PBS containing Kolliphor^®^ HS 15 (0.06 mg/mL) for SPECT/CT imaging, and with 5 MBq/0.5 nmol for ex vivo biodistribution studies (*n* = 4 mice/group).

#### 4.5.1. SPECT/CT Imaging

The small-animal VECTor^5^/CT (MILAbs B. V.; Utrecht, The Netherlands) was used for all imaging studies. Image acquisition was performed at 4, 24, 48, and 72 h post-injection using the high-energy general-purpose mouse collimator (HE-GP-M, 0.8 mm pinhole size) in list mode. Corresponding CT scans were acquired in total-body and normal mode (50 kV, 0.21 mA, 75 ms) for anatomical reference and attenuation correction. All SPECT images were reconstructed using the similarity-regulated ordered-subsets expectation maximization (SROSEM) algorithm (MILAbs Rec 11.00 software, MILAbs, Utrecht, The Netherlands) with 5 iterations, 128 subsets, and a voxel size of 0.4 mm^3^. Image processing and analyses of the reconstructed data were performed using the PMOD image analysis software version 3.10 (PMOD Technologies; Zurich, Switzerland) to calculate the percentage of injected dose per mL (% ID/mL). To allow quantification of the SPECT data, calibration factors were derived from [^177^Lu]Lu phantoms.

#### 4.5.2. Ex Vivo Biodistribution

For the ex vivo biodistribution studies, animals were euthanized at selected time points (4, 24, 48, and 72 h) after injection. Specific tissues and tumors were excised, and their radioactivity uptake was determined. The following organs were collected from each animal: blood, tumor, heart, lung, liver, spleen, stomach, intestine, pancreas, kidney, muscle, skin, bone, and bone marrow. To confirm receptor specificity, mice were co-injected with [^177^Lu]Lu-JR11 or [^177^Lu]Lu-**8a** and a 50-molar excess of their respective unlabeled compound (JR11 or **8a**), after which uptake in organs and tumor was determined at 24 h post-injection. All tissues were weighed and counted in a *γ*-counter, and data were reported as percentage injected dose per gram of tissue (% ID/g).

#### 4.5.3. Statistical Analysis

Statistical analysis and nonlinear regression were performed using GraphPad Prism 9 (GraphPad software, San Diego, CA, USA), and a Mann–Whitney test was used to compare medians between groups. Data were reported as mean ± SEM (standard error of mean) for at least three independent replicates.

## 5. Conclusions

Our manuscript reported a successful synthesis of two long-acting JR11 analogs for improved radionuclide therapy of NETs. Radiolabeling of both analogs with lutetium-177 was achieved with very high RCYs and RCPs. Both radiopeptides showed excellent stability in PBS and mouse serum, conserved their hydrophilic behavior, and exhibited good binding to human albumin. Compounds **8a** and **8b** showed good binding affinity towards SSTR2, high cell uptake, and low internalization rate. [^177^Lu]Lu-**8a** demonstrated extended residence in the blood, higher kidney uptake, and nonspecific tumor accumulation compared to [^177^Lu]Lu-JR11. Unfortunately, [^177^Lu]Lu-**8b** did not show any tumor uptake despite the high potential of the 4-(*p*-methoxyphenyl)butyryl ABD. Although insertion of a 4-(*p*-iodophenyl)butyryl ABD into JR11 improved its blood circulation, as expected, we also noticed high uptake in non-target organs with [^177^Lu]Lu-**8a**. Therefore, further optimization is required to combine an ABD and JR11 to obtain a long-acting SSTR2 antagonist with an adequate biodistribution and pharmacokinetic profile for safe and efficient radionuclide therapy of neuroendocrine tumors.

## Data Availability

Data is contained within the article and Appendix A.

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
