# Peer review of "Synthesis and Evaluation of Two Long-Acting SSTR2 Antagonists for Radionuclide Therapy of Neuroendocrine Tumors"

_pharmaceuticals, 2022, doi:10.3390/ph15091155_

Round 1

Reviewer 1 Report

This paper describes the development of two long-acting SSTR2 antagonists for a potential application in radionuclide therapy of neuroendocrine tumors. More precisely, this work focused on systems designed on JR11 peptide, a well-known SSTR2 antagonist, associated to a DOTA macrocycle as chelating agent for lutetium-177. The newness is illustrated by the addition of a lysine residue as linker in order to introduce an albumin binding domain (ABD). The main idea of this approach being to extend blood circulation time to promote tumor uptake and increase received dose.

For that purpose, two peptides derivatives were prepared by solid-phase peptide synthesis, with conjugation on the additional lysine of two moieties 4-(p-iodophenyl)butyryl (compound 8a) or 4-(p-methoxyphenyl)butyryl (compound 8b), both already described as ABD. In addition to high RCYs and high RCPs, good stability in PBS and mouse serum were reported for both compounds radiolabeled with lutetium-177. Even if these two compounds showed an improved binding to human albumin, values for logD7.4 and IC50 are higher than for the reference [177Lu]Lu-JR11 suggesting that peptide modification interfered in the lipophilicity and affinity of the final compound. This tendency is confirmed by in vitro assays showing a lower uptake and internalization in cell lines expressing SSTR2. In vivo evaluation by SPECT/CT imaging revealed a tumor uptake for compound 8a and surprisingly none for compound 8b, however, a higher and longer accumulation in kidneys was measured for both radiotracers in regard to [177Lu]Lu-JR11. Finally, biodistribution study of compound 8a validated observations made with imaging, but data from a comparison with a block group that received the non-radioactive analog revealed a non-specific accumulation in tumor for compound 8a. This could be explained by an increase of the global uptake in all organs due to the prolonged blood circulation time of the compound, inducing a loss of specificity toward tumor cells. These results indicate that in that case, addition of these ABD as well as peptide modifications are not appropriate and that further experiments are needed for a use of this approach in SSTR2 targeting.

The manuscript is comprehensive, structured clearly and logically, furthermore, the work presented is significant and very well discussed. The general strategy is easy to follow, results are carefully reported and the discussion is consistent and detailed. Besides, I have no specific comment to add about language, grammar or nomenclature used in the manuscript.

I really appreciated this work, and even if results are may be different of what it could be expected, I believe this study can serve as basis for an improvement of this approach and that is why I recommend a publication of this article in its current form.

Author Response

Dear reviewer, we really appreciate your comments and would like to thank you for your positive feedback and your interest in our work. As you mentioned, we hope that this manuscript is the basis for future work on improving the pharmacokinetics of JR11.

Reviewer 2 Report

After reading the manuscript, I don't think I have to background to comment or critique the methods. 

The background provided a clear background and review of the literature of why this project was undertaken.  The discussion provided a clear description of the relevance and limitations of this work. 

Author Response

Dear reviewer, thank you very much for the time you have taken to review the article. We recognize your valuable feedback.

Reviewer 3 Report

This manuscript presents the development of two SSTR-2 antagonists to improve the pharmacokinetic profile of JR11. The authors have incorporated the previously reported two albumin binding domains (ABD), namely 4-(p-iodophenyl)butyryl and 4-(p-methoxy-phenyl)butyryl, into the peptide sequence of JR11 in order to extend blood circulation time. The synthesis of JR11 analogs 8a and 8b was carried out using standard solid phase peptide synthesis (SSPS), followed by radiolabeling with 177LuCl3. The newly synthesized 8a exhibited stronger interaction with plasma proteins and higher uptake in U2OS-SSTR2 cells compared with 8b. Although the introduction of ABD into JR11 improved its blood residence time and higher uptake by tumor cells, this modification had a negative impact on the selectivity toward tumor versus normal cells. Compound 8a significantly accumulated in other organs including the heart, lung, liver, and pancreas, which appears to be a major issue. Overall, the experiments were carefully designed and carried out. This is an interesting study and merits publication after addressing the comments below. 

1. Have the authors tried alternate approaches to improve the pharmacokinetics profile? For example, structural modifications in JR11 in order to slow the drug metabolism rather than binding to plasma proteins?

2. Despite knowing that albumin binding moieties can negatively affect the pharmacokinetic profile of peptides, why did the authors choose this approach?

3. Why [177Lu]Lu-8b didn’t show tumor uptake?

4. Line 68 “Coupling of the chelator was performed using PyBOP, as we previously observed faster reaction kinetics when using this coupling agent.” Please include the reference. 

5. Scheme 1- the authors have mentioned that 2,5-dioxopyrrolidin-1-yl 4-(p-tolyl)butanoate was used for 8b preparation. However, 4-methoxyphenyl butanoate is shown in compound 8b. This discrepancy needs to be corrected.

6. Why is [177Lu]Lu-8a bound more efficiently to human albumin than [177Lu]Lu-8b? 

Author Response

Dear reviewer, 

We would like to thank you for the time you have taken to review our manuscript. We really appreciated your comments and suggestions concerning our paper and we hope that our answers will meet your expectations.

The revised draft of the manuscript can be found in the track changes mode in Microsoft Word version

Our comments and answers will be given in blue below:

1. Have the authors tried alternate approaches to improve the pharmacokinetics profile? For example, structural modifications in JR11 in order to slow the drug metabolism rather than binding to plasma proteins?

We effectively tried other approaches to slow down the metabolism and consequently improve the bioavailability of JR11 by modifying the chemical linkage during the cyclization of the peptide. Few chemical strategies were explored to replace the native disulfide bound, supposedly being metabolically fragile, but none of the compounds synthesized exhibited a good binding affinity for SSTR2 and it was decided to not perform further biological evaluation with these JR11 derivatives. Furthermore, it is also known that JR11 is very sensitive to chemical modifications, especially at the N-terminus, and structural modifications of the peptide sequence can significantly affect its binding properties. Thus, we opted for the introduction of an albumin binding domain to modify the pharmacokinetic properties of JR11.

2. Despite knowing that albumin binding moieties can negatively affect the pharmacokinetic profile of peptides, why did the authors choose this approach?

We agree that addition of an albumin binder affects the pharmacokinetic profile of peptides, but our main objective was to increase the blood half-life of JR11, as it is known that more than two third of the radiolabeled JR11 is cleared from the blood within few minutes after administration. Therefore, addition of an albumin binding moiety to JR11 could extend its blood circulation and ultimately lead to more tumor accumulation, which was somehow confirmed by the data obtained for [177Lu]Lu-8a.

3. Why [177Lu]Lu-8b didn’t show tumor uptake?

Unfortunately, we don’t have a clear explanation about the absence of tumor uptake of [177Lu]Lu-8b despite its affinity for SSTR2.

4. Line 68 “Coupling of the chelator was performed using PyBOP, as we previously observed faster reaction kinetics when using this coupling agent.” Please include the reference. 

As you suggested, two references were included (line 70, ref. 15 & 16).

5. Scheme 1- the authors have mentioned that 2,5-dioxopyrrolidin-1-yl 4-(p-tolyl)butanoate was used for 8b preparation. However, 4-methoxyphenyl butanoate is shown in compound 8b. This discrepancy needs to be corrected.

Changes were done line 84 and 380.

6. Why is [177Lu]Lu-8a bound more efficiently to human albumin than [177Lu]Lu-8b? 

A brief explanation was mentioned in our manuscript about this point (line 228). However, more details were provided in the supplemental information of the work published by Hsiou-Ting Kuo et al. (ref. 13, Table 8 in SI). In fact, it appears that the retention of the radiotracer by the plasma proteins is modulated by the lipophilicity of the benzyl group. Kuo and coworkers actually reported that the average blood retention of their PSMA derivatives at 1 and 3 h after injection generally followed the lipophilicity order of the substituents (Fig. 3 in SI), with the exception of the F-substituted derivative.